# Targeting Mitochondrial Metabolic Reprogramming as a Potential Approach for Cancer Therapy

**DOI:** 10.3390/ijms24054954

**Published:** 2023-03-04

**Authors:** Liufeng Zhang, Yuancheng Wei, Shengtao Yuan, Li Sun

**Affiliations:** Jiangsu Key Laboratory of Drug Screening, China Pharmaceutical University, Nanjing 211198, China

**Keywords:** cancer, mitochondria, metabolic reprogramming, drug development

## Abstract

Abnormal energy metabolism is a characteristic of tumor cells, and mitochondria are important components of tumor metabolic reprogramming. Mitochondria have gradually received the attention of scientists due to their important functions, such as providing chemical energy, producing substrates for tumor anabolism, controlling REDOX and calcium homeostasis, participating in the regulation of transcription, and controlling cell death. Based on the concept of reprogramming mitochondrial metabolism, a range of drugs have been developed to target the mitochondria. In this review, we discuss the current progress in mitochondrial metabolic reprogramming and summarized the corresponding treatment options. Finally, we propose mitochondrial inner membrane transporters as new and feasible therapeutic targets.

## 1. Introduction

Mitochondria, as important organelles in cells, produce ATP through oxidative phosphorylation [1]. In addition, mitochondria perform many other biological functions, including producing reactive oxygen species (ROS), reduction-oxidation (REDOX) molecules and metabolites, participating in anabolic metabolism, and regulating cell signaling and cell death [2]. Previous studies have shown that cancer is a mitochondrial metabolic disease [3]. The Warburg effect makes aerobic glycolysis the main source of energy for cancer cells [4], but new research results show that mitochondria in malignant cells are still active and closely related to the occurrence of cancer [5]. The role of mitochondria in the context of metabolic reprogramming is gradually being revealed, which has led to an increasing focus on targeted mitochondrial therapy.

In this review, we summarize recent progress in our understanding of mitochondrial involvement in metabolic reprogramming. We describe the different targets and small molecule compounds that have been developed for mitochondria, summarize the characteristics of the various research and development ideas, and discuss the research and development difficulties caused by the high impenetrability of the inner mitochondrial membrane. Finally, we propose and collate therapeutic strategies that use mitochondrial intima transporters as therapeutic targets to enhance the efficacy of targeted mitochondria.

## 2. Mitochondrial Metabolic Reprogramming Promotes Tumor Progression

Routine oxygen and nutrient supplies cannot meet the demands of cancer cells in solid tumors, which leads to significant metabolic stress [6] and necessitates metabolic reprogramming. It is widely accepted that metabolic reprogramming is a hallmark of cancer. Metabolic reprogramming is not stable and the interaction between several oncoproteins and tumor suppressors changes cellular metabolic pathways, thereby promoting tumor progression. Metabolic reprogramming alters the use of carbohydrates, lipids, and amino acids in cancer cells [6].

### 2.1. Glucose Metabolic Reprogramming

Glucose is the main carbon source and energy for cell growth and proliferation. Glucose undergoes three main pathways for energy conversion, including aerobic oxidation, glycolysis, and pentose phosphate pathway. Glucose metabolism is significantly different in cancer cells. Warburg effect is the dominant method of glucose metabolism in cancer cells. Tumor cells absorb a large amount of glucose by overexpressing glucose transporter 1 (GLUT1). Glucose is eventually converted into pyruvate, as the end product of aerobic glycolysis [7]. Subsequently, pyruvate is mostly converted into lactate. Lactate is then transported to the extracellular space by a monocarboxylate transporter 4 (MCT4) transporter and acidifies the tumor microenvironment [8]. Mitochondrial pyruvate carrier (MPC) transports the remaining pyruvate to mitochondria, where it is converted into Acetyl-CoA as a substrate for the tricarboxylic acid (TCA) cycle. Transcriptional activation of mitochondrial pyruvate dehydrogenase kinase 1 inactivates pyruvate dehydrogenase (PDH), ultimately preventing Acetyl-CoA production in mitochondria in tumor cells [9]. In addition, tumor cells overexpress pyruvate kinase M2, thereby blocking the conversion of phosphoenolpyruvate to pyruvate and enhancing other biosynthetic pathways to promote tumor cell proliferation [10].

The Warburg effect also increases nicotinamide adenine dinucleotide (NADH) generation and activates the pentose phosphate pathway to provide nicotinamide adenine dinucleotide phosphate (NADPH), which maintains REDOX homeostasis of cancer cells [11]. NADPH is used for reducing reactions in the synthesis of biomolecules, such as fatty acids, cholesterol, deoxyribose, tetrahydrofolate, and other substances. NADPH is also needed for the reduction of oxidized glutathione to maintain the REDOX balance in cancer cells [12]. Therefore, the abnormal pentose phosphate pathway has also become a hot spot in cancer research.

Metabolic reprogramming of glucose implies that cancer cells bypass mitochondrial respiration and use aerobic glycolysis for energy supply. The Warburg effect explains glycolysis under aerobic conditions. Recent findings suggest that saturation of NADH shuttling, but not the need for cancer cell proliferation, facilitates aerobic glycolysis [13]. Increased mitochondrial respiration in breast tumor cells after lactate dehydrogenase A (LDHA) inhibition suggests that cancer cells retain the ability to oxidize glucose through oxidative phosphorylation (OXPHOS) in their mitochondria [14]. Glucose oxidation by the TCA cycle in tumor cells in isotope tracing experiments also confirmed the above thesis [15].

### 2.2. Lipid Metabolic Reprogramming

Lipid metabolic reprogramming mainly affects fatty acid biosynthesis, oxidation, intake, and modification [16]. Cellular lipids are mainly composed of fatty acids, triglycerides, sphingolipids, phospholipids, and cholesterol. Most lipids are derived from fatty acids. Lipids are the building blocks of cell membranes, second messengers, and cellular energy sources. In normal cells, extracellular uptake of lipids is the main pathway of providing cellular lipids, while in cancer cells, the PI3K/Akt signaling pathway upregulates enzymes required for fatty acid synthesis [17]. Increased de novo synthesis of fatty acids changes the composition of intracellular lipids. For instance, it decreases polyunsaturated fatty acid (PUFA) and increases monounsaturated fatty acid (MUFA). Overproduction of MUFA attenuates the damage caused by PUFA peroxidation in the presence of ROS [18].

Mitochondria are deeply involved in lipid reprogramming in tumor cells. As an intermediary for the TCA cycle, Acetyl-CoA is involved in the regulation of lipid metabolism in cancer cells [19]. Acetyl-CoA is an important substrate for fatty acid synthesis. Acetyl-CoA is regulated by three intracellular enzymes, including acyl-CoA synthetase short-chain family member 2 (ACSS2), ATP citrate lyase (ACLY), and Acetyl-CoA carboxylase (ACC). Furthermore, ACSS2 catalyzes the conversion of extracellular acetic acid to Acetyl-CoA, which is highly consumed by cancer cells [20]. Mitochondrial citric acid is transported to the cytoplasm via SLC25A1 and converted to Acetyl-CoA by ACYL [21]. On the other hand, ACC converts Acetyl-CoA into malonyl CoA during fatty acid synthesis [22]. ACLY connects lipid and glucose metabolism in tumor cells and forms a complex metabolic network.

### 2.3. Amino Acid Metabolic Reprogramming

Tumor cells immensely need amino acids for protein synthesis. As an important component of tumor metabolic reprogramming, glutamine metabolic reprogramming plays an important role in maintaining tumor cell energy homeostasis, ROS balance, and the continuous activation of mTOR [23]. Some tumor cells consume large amounts of glutamine to meet their metabolic needs. Extracellular glutamine intake provides carbon and nitrogen for anabolism and energy production. Intracellular deamination of glutamine and its conversion into glutamate is catalyzed by phosphate-dependent glutaminases (GLS1 and GLS2) [24]. Glutamate is further catabolized through the TCA cycle. It is catalyzed to α-ketoglutaric acid (α-KG) by glutamate dehydrogenase (GDH) or aminotransferase or directly by glutathione cysteine ligase. Glutathione synthetase catalyzes the formation of glutathione (GSH) to maintain intracellular REDOX homeostasis. αKG enhances the TCA cycle and maintains mitochondrial integrity and activity through succinyl-CoA oxidation and nicotinamide adenine dinucleotide (NAD^+^) reduction [25].

The abnormal expression of regulatory factors in tumor cells often affects glutamine metabolism. These regulators are usually oncogenes and tumor suppressor genes. Among them, Myc binds to the promoter elements of glutamine transporters (such as SLC7A5 and SLC1A5) to induce glutamine transportation [26]. When KRAS is activated, the GOT2-GOT1-ME1 pathway can overexpress genes involved in glutamine decomposition [27]. Therefore, glutamine becomes the main carbon source for the TCA cycle. In addition, by upregulating NRF2, KRAS induces the NRF2-mediated antioxidant system to maintain REDOX balance and promote tumorigenesis [28,29] (Figure 1).

## 3. Targeting Mitochondrial Metabolism for the Therapy of Cancers

As we mentioned above, mitochondrial metabolic reprogramming supports the development and progression of tumors. Given the importance in cancer cells, mitochondrial metabolism can be a promising point for the development of anti-tumor drugs [30]. Accordingly, we focus on TCA cycle, OXPHOS, ROS and mtDNA to understand targeting mitochondrial for the therapy of cancers.

### 3.1. Targeting TCA Cycle

Many studies have shown that the Warburg effect is pivotal for the development and progression of tumors. However, recent studies have shown that the role of the TCA cycle and OXPHOS in tumor metabolism cannot be ignored [31,32]. The TCA cycle refers to Acetyl-CoA oxidization to H_2_O and CO_2_. The TCA cycle occurs in the mitochondria, which are the final metabolic pathways for carbohydrates, lipids, and amino acids.

Abnormal levels of isocitrate dehydrogenase (IDH) and succinate dehydrogenase (SDH) can lead to abnormal function of the TCA cycle, which may be related to tumorigenesis [33]. IDH has three subtypes, including IDH1, IDH2, and IDH3. IDH catalyzes the oxidation and carboxylation of isocitrate to produce α-KG. Mutations in IDH1 and IDH2 genes lead to increased production of D-2 hydroxy-glutarate (2HG), contributing to the development of various malignancies such as acute myeloid leukemia, chondrosarcoma, cholangiocarcinoma, and glioma [34,35,36]. IDH Mutations are associated with highly heterogeneous tumor microenvironments, suggesting that targeting IDH mutations may effectively treat cancer [37]. The novel role of IDH in various malignant tumors has led to the development of IDH inhibitors. Enasidenib and ivosidenib are approved IDH inhibitors, which have shown significant clinical benefits in acute myeloid leukemia and refractory cholangiocarcinoma [38,39,40].

SDH consists of four subunits, including SDHA, SDHB, SDHC, and SDHD. It catalyzes the conversion of succinic acid to fumarate. SDH deletion has been found in gastrointestinal stromal tumors and paragangliomas [41]. Studies have shown that the downregulation of SDHC promotes epithelial-mesenchymal transition (EMT) and is accompanied by structural remodeling of mitochondria. SDHC downregulation is also associated with malignant progression, tumor heterogeneity, and drug resistance [42]. Therefore, targeted SDH can potentially treat cancer. Tumor necrosis factor receptor-associated protein 1 (TRAP1) is a mitochondrial chaperone protein belonging to the heat shock protein 90 (HSP90) family. It is highly expressed in many types of tumors. TRAP1 inhibits mitochondrial complex II, downregulates SDH activity, and promotes tumor growth [43,44]. In addition to pseudohypoxia, TRAP1 protects tumor cells from oxidative stress [43].

Several TRAP1 inhibitors are already available. Gamitrinib, a small-molecule TRAP1 and HSP90 inhibitor, is currently passing its phase I clinical trial in patients with advanced cancer, and the results of animal experiments show that gamitrinib is a safe and effective anticancer therapy [45]. The mitochondrial osmotic drug, DN401, is a newly discovered pan-inhibitor of HSP90 that inhibits the HSP90 family, including TRAP1. It has stronger anticancer activity than other HSP90 inhibitors [46]. Honokiol bis-dichloroacetate (HDCA) is a small-molecule compound that specifically targets TRAP1. HDCA can restore the TRAP1-dependent downregulation of SDH, reduce tumor cell proliferation, increase mitochondrial superoxide levels, and inhibit tumor growth by selective inhibition of TRAP1 [47].

P53, a well-known tumor suppressor, can inhibit the expression of pyruvate dehydrogenase kinase 2 (PDK2), thus activating the oxidative metabolism of mitochondria and promoting the TCA cycle [48]. In addition, p53 can also induce mitochondrial GLS2 expression to enhance GSH synthesis and α-KG, thus promoting the TCA cycle [49].

P53 function is often impaired in tumors. Murine double minute 2 (Mdm2) and murine double minute X (MdmX) are major negative regulators of P53. They can independently or together inhibit p53 [50]. Blocking Mdm2 and MdmX is a potential strategy for treating tumors. Idasanutlin (RG7388) is a small-molecule Mdm2 inhibitor currently in phase III of trials. In vivo results showed that RG7388 effectively reduced cell proliferation and induced p53-dependent pathways, cell cycle arrest, and apoptosis, thereby inhibiting tumor growth [51]. Milademetan is also a small-molecule inhibitor of Mdm2. It has been used in the clinical trials of advanced solid tumors and hematological malignancies, such as liposarcoma and acute myeloid leukemia [52]. ALRN-6924 is a dual-target inhibitor of Mdm2/MdmX, which has been used in a phase I clinical trial. ALRN-6924 stably activates p53-dependent transcription at single-cell and single-molecule levels. It exhibited biochemical and molecular targeting activity in in vitro and in vivo studies of leukemia [53]. Current clinical trials have shown that ALRN-6924 is well tolerated and has antitumor activity in patients with solid tumors or lymphomas carrying wild-type TP53 [54]. FL118 is a camptocamptoid analog. FL118 can change the targeting specificity of the Mdm2-MdmX E3 complex from p53 to MdmX, thereby accelerating MdmX degradation and activating p53-dependent aging in colorectal cancer (CRC) cells [55]. The combination of FL118 and cisplatin can synergically kill drug-resistant pancreatic cancer cells, prevent the globular formation of treatable pancreatic cancer stem cells, and overcome chemoresistance [56].

### 3.2. Targeting OXPHOS

OXPHOS also plays an important role in tumor metabolism. OXPHOS converts oxygen to water and simultaneously releases energy for ATP production. Studies have shown that OXPHOS can provide ATP for tumor proliferation [57]. The electron transport chain (ETC) is an important component of OXPHOS. ETC is composed of complex I-IV, CoQ, and Cyt c.

ETC is necessary for tumor growth. Mitochondrial complexes I and II transfer electrons to panquinone, resulting in panthenol production. Complex III oxidizes panthenol to panquinone. The absence of mitochondrial complex III impairs tumor growth. Tumor growth requires the ETC-mediated oxidation of panthenol [31]. Several types of tumors, including CRC, ovarian cancer, acute myeloid leukemia, and glioblastoma, have somatic mtDNA mutations of complex I, III, or IV [58].

In addition to ATP, mitochondrial respiration also produces biosynthetic precursors such as aspartic acid. ETC inhibition promotes activating transcription factor 4 (ATF4) and mTORC1 signaling pathways by consuming aspartic acid and asparagine. The combination of ETC inhibitor and restriction of aspartic acid impaired tumor growth in animal models [59].

Complex I is located at the frontline of the respiratory chain. As a major producer of proton gradients in ETC, complex I is a suitable target for the development of an OXPHOS inhibitor. As a marketed drug, metformin has received much attention for its ability to inhibit complex I, but its low potency has limited the potential for re-purposing [60]. BAY87-2243 was also promising, but severe vomiting in phase I trials prevented its further development. In recent years, more and more efficient and selective small-molecule drugs have been developed [61,62]. EVT-701, a novel small-molecule inhibitor targeting diffuse large B-cell lymphoma and NSCLC, has demonstrated good efficacy in vitro and in vivo. It should be emphasized that the original structure of BAY87-2243 was deliberately modified to prevent severe side effects in clinical trials [63]. Furthermore, Kazuki Heishima et al. found that the plant extract, petasin (PT), is a complex I inhibitor that mainly inhibits tumor growth in animal models, with high efficiency and low toxicity [64]. In addition, human epidermal growth factor receptor 2 (ERBB2) inhibitor, mubritinib, has anticancer properties by inhibiting complex I [65].

The steroid saponin Gracillin is a natural compound with potent antitumor activity. Its antitumor property depends on mitochondrial complex II inhibition [66]. In vitro studies have shown that Gracillin inhibits mitochondrial complex II-mediated energy production, thereby reducing the viability and colony formation of breast cancer cells. It has shown marked antitumor activity in animal models [67]. Atovaquone is an FDA-approved drug for the treatment of malaria. It is also a potent selective OXPHOS inhibitor with anticancer properties through mitochondrial complex III inhibition [68]. Arvinder et al. showed that atovaquone can inhibit ovarian cancer cell proliferation and growth in vitro and in vivo [69]. Two complex IV inhibitors have been approved for cancer treatment: mitotane for adrenocortical cancer, and arsenic trioxide for acute promyelocytic leukemia. Arsenic trioxide has shown promising results in the preclinical studies of glioblastoma (GBM) [70]. In addition, loss of BTB and CNC homology1 (BACH1) has been reported to increase the sensitivity of some types of cancer to ETC inhibitors (such as metformin). It indicates that combined inhibition of ETC and BACH1 may effectively treat cancer [66,71].

### 3.3. Targeting ROS

Electron leakage in ETC results in the conversion of O_2_ to O_2_^−^ in the mitochondrial matrix. O_2_^−^ is one of the ROS with strong oxidizing properties. Normal cells use antioxidant enzymes (catalase, peroxidase, glutathione peroxidase, and superoxide dismutase) and small molecules of antioxidants (vitamin C, vitamin E, and beta-carotene) to remove ROS. Cancer cells produce high levels of ROS, which disrupt REDOX homeostasis and activate many oncogenic signaling pathways [72,73]. On the other hand, ROS also exert anticancer activity. The absence of cysteine desulfurase (NFS1) significantly enhances the sensitivity of CRC cells to oxaliplatin. It induces apoptosis, necrosis, pyroptosis, and iron-mediated cell death by increasing ROS levels. High expression of NFS1 is likely related to MYC [74].

A series of compounds have been discovered to inhibit ROS. Lexibulin blocks ROS production and inhibits tumor growth through endoplasmic reticulum (ER) stress [75]. Bavachin can cause ROS accumulation and induce iron-mediated death via the STAT3/P53/SLC7A11 axis [76]. Darinaparsin, an organic arsenic molecule with anti-cancer activity, was approved in Japan in June 2022. It can induce G2/M cell cycle arrest and apoptosis in tumor cells by disrupting mitochondrial function and increasing ROS production [77]. Curcin C, a type I ribosome-inactivating protein, can increase ROS levels in osteosarcoma cells, alter mitochondrial membrane potential, and weaken the antioxidant system, thereby inhibiting the proliferation of various osteosarcoma cell lines [78]. Auriculasin also exerts anticancer effects through ROS. Auriculasin induces ROS production in a concentration-dependent manner to promote CRC cell apoptosis and iron-mediated death [79].

### 3.4. Targeting mtDNA

As a multi-copy genome, mtDNA is highly mutated and plays an important role in tumorigenesis and tumor progression [80]. The development of mitochondrial genome tools has markedly advanced research on mtDNA mutations. For example, mutations of the MT-ATP6 and MT-ND5 genes have been shown to increase ROS levels to promote tumor growth. Mutated mtDNA also enhances metastasis via ROS [81,82]. CCC-021-TPP was developed as a small molecule compound targeting ND6 A14582G mutation in mtDNA of non-small cell lung cancer. CCC-021-TPP can increase mitochondrial ROS production and induce mitochondrial autophagy to inhibit cancer progression [83]. Huang et al. designed and synthesized N-(N′,N′-diethanolaminopropyl) benzothiophenonaphthalimide (7C). They found that this compound effectively induces an mtDNA sequence named HRCC, thereby reducing mitochondrial membrane potential, increasing ROS production, and inhibiting tumor growth [84].

### 3.5. Prospective and Limitations

Inhibition of mitochondrial metabolism can effectively prevent tumor progression. However, there are several challenges. For example, ROS is a double-edged sword for cancer survival [85]. Although the accumulation of an excessive amount of ROS can undermine cancer cell survival, DNA damage caused by high levels of free radicals and genomic instability increase the risk of cancer [86]. In addition, inhibition of ROS alone has no significant effect on tumor growth [87], which means that extreme control of ROS production cannot be therapeutic. Developing compounds that target the p53 pathway is also extremely challenging. How p53 prevents tumor development is still unclear, and most tumor cells contain at least a dysfunctional tp53, making it an undruggable target [88]. Recent clinical studies have shown that targeting OXPHOS is not very feasible. IACS-010759, a small molecule inhibitor targeting complex I, had a narrow therapeutic index and serious side effects such as elevated lactic acid level and neurotoxicity in a phase I clinical study [89]. Therefore, it is necessary to monitor the toxicity of anti-tumor drugs targeting complex I.

It is worth noting that the specific structure of mitochondria itself is also a source of difficulty in drug development. From the outside to the inside, mitochondria can be divided into five functional regions: outer mitochondrial membrane (OMM), intermembrane space, inner mitochondrial membrane (IMM), the cristae space and the mitochondrial matrix [90]. Even though the outer mitochondrial membrane appears to be highly permeable, when confronted with the inner mitochondrial membrane, there is tremendous difficulty for molecules. Therefore, the development of compounds that target intracellular mitochondrial membrane transporters is emerging as an efficient and effective option (Figure 2).

## 4. Inner Mitochondrial Membrane Transporters: The Significative Research and Development Direction

Mitochondria utilize a variety of highly specific transporters to support molecules exchange. Currently, transporters located in the inner mitochondrial membrane mainly include members of the families SLC25, SLC56, SLC1 and MPC. Although the transport substrates vary between families and their members, the abnormal expression of several transporters has been found in cancer to varying degrees, making it possible to target mitochondrial endo-membrane transporters.

### 4.1. SLC25 Family

The SLC25 family (mitochondrial carrier family, MCF) consists of 53 members and is the largest solute transporter family in humans [91]. All carriers possess a tripartite structure with three tandem repeats of homologous domains of about 100 amino acids. Each repeat contains two hydrophobic stretches [92]. MCs have six trans-membrane helices, with the N- terminal and C-terminal located on the cytosolic side. Three replicates are connected by two loops on the cytosolic side, and two transmembrane α-helices in each replicate are connected by three loops on the matrix side [93]. Based on substrate specificity, MCs are classified into four groups: amino acids carriers, nucleotides and dinucleotide carriers, carboxylates and keto acids carriers, and carriers of additional substrates [94]. Previous studies have revealed the close relationship between SLC25 and carbon sources in cancer cells.

As the source of Acetyl-CoA, citrate plays a significant role in fatty acid synthesis in the cytoplasm. Meanwhile, citrate is involved in the Krebs cycle and oxidative phosphorylation in mitochondria [95]. SLC25A1, also named mitochondrial citrate/isocitrate carrier (CIC), is the only known transporter located in the inner mitochondrial membrane. CIC exchanges mitochondrial and cytosolic citrate [96]. Previous studies reported the overexpression of CIC in several types of cancers. In CRC, PPARγ co-activator 1α (PGC1α) upregulated CIC expression. In addition, low expression of SLC25A1 significantly inhibited the growth of CRC cells by inhibiting G1/S cell cycle progression and inducing apoptosis [97]. CIC enhanced de novo lipid synthesis and upregulated OXPHOS to sustain the survival of CRC cells [98]. In non-small cell lung cancer (NSCLC), CIC is highly expressed at metastatic sites or during acute and chronic hypoxia [99]. CIC enables CSCs to use citric acid for mitochondrial respiration and mitigates the deleterious effects of ROS produced by the activation of the IDH2-NADPH system [96].

SLC25A8, also known as mitochondrial uncoupling protein 2 (UCP2), uses four-carbon metabolites and Ca^2+^ as its substrates. UCP2 translocates mitochondrial glutamine-derived aspartate into the cytoplasm and supports pancreatic cancer growth [100]. Mitochondrial ROS are the main source of intracellular oxidative stress. ROS can damage biomolecules, induce the conversion of guanine to 8-oxo-guanine, oxidize amino acids, cysteine, and methionine, and facilitate lipid peroxidation [101,102,103]. UCP2 maintains ROS at acceptable levels and prevents mitochondrial and cellular dysfunction [104]. It plays a contradictory role in tumorigenesis. Loss of UCP2 leads to metabolic reprogramming of colonic cells and impairs REDOX homeostasis, facilitating malignant transformation [105]. On the other hand, UCP2 overexpression is observed in advanced tumors and is associated with decreased survival. UCP2 overexpression is thought to be associated with the Warburg effect [106,107]. UCP2 catalyzes the exchange of malate, oxaloacetic acid, and aspartate with phosphate and exports mitochondrial C4 metabolites to the cytoplasm [108]. Ectopic expression of UCP2 in cancer cells leads to the metabolic transition from mitochondrial oxidative phosphorylation to glycolysis [106].

SLC25A10, also known as dicarboxylate carrier, plays an important role in energy metabolism and REDOX homeostasis. It mainly transports malate, phosphate, succinate, sulfate, and thiosulfate, providing substrates for sulfur metabolism and gluconeogenesis. SLC25A10 is highly expressed in CRC, human osteosarcoma, ovarian cancer, and lung cancer. Knockdown of SLC25A10 can significantly inhibit the proliferation of cancer cells and increase their glutamine dependence [109,110,111,112].

SLC25A11 encodes the carrier of oxglutaric acid, and SLC25A12 encodes the carrier of aspartic acid-glutamate 1 [92,113]. Both of them are located in the inner mitochondrial membrane and together constitute the malate-aspartic acid shuttle system. Oxyglutaric acid carriers transport cytoplasmic malic acid to the mitochondrial matrix and export α-ketoglutaric acid from the mitochondrial matrix to the cytoplasm. Malate dehydrogenase converts mitochondrial malic acid into oxaloacetic acid. Then, aspartic aminotransferase converts oxaloacetic acid to aspartic acid, which will finally be transported to the cytoplasm by the aspartic acid-glutamate carrier 1 [114]. Due to the less permeable of IMM to NADH [115], malate dehydrogenase catalyzes the oxidation of malate to oxaloacetate and simultaneously reduces mitochondrial NAD^+^ to NADH/H^+^ to produce ATP. The malate-aspartic acid shuttle is essential for glycolysis. The expression level of SLC25A11 is closely related to cancer. The expression level of SLC25A11 in normal alveolar epithelial cells and inflammation transiently upregulates SLC25A11 in non-cancerous epithelial cells. However, the expression of SLC25A11 is much higher in NSCLC tissues than in normal lung tissue [116]. SLC25A11 increases mitochondrial membrane potential in cancer cells. Mitochondria rely on the malate-aspartate NADH shuttle (MAS) system for NADH transportation and ATP production. SLC25A11 knockdown significantly reduced ATP production and inhibited cancer cell proliferation by blocking the mTOR phosphorylation and downregulation of c-Myc and eIF4B [116]. SLC25A12 overexpression in liver cancer is associated with poor prognosis. Acetylated histones promote the expression of SLC25A12 by regulating cAMP response element-binding protein (CREB) function. Silencing of SLC25A12 resulted in G1/S cycle arrest of HepG2 cells and significantly impaired their proliferation [117]. However, low expression of SLC25A12 has been identified as a contributing factor to lung metastasis due to the deregulated folate pathway in aspartate/glutamate carrier 1 (AGC1) deficiency [118].

In addition, Zhang et al. indicated that the abnormal overexpression of SLC25A29 in cancer helps arginine transportation into mitochondria and upregulates mitochondrial NO, thereby suppressing mitochondrial respiration, enhancing glycolysis, and promoting cancer progression [119]. SLC25A51 is a newly identified mammalian mitochondrial NAD^+^ transporter [120]. Previous studies have confirmed that SLC25A51 is significantly overexpressed in human hepatocellular carcinoma (HCC) and enhances glycolysis and HCC progression by activating sirtuin 5 (SIRT5) [121]. SLC25A18 transports glutamate through the inner mitochondrial membrane. Increased expression of SLC25A18 inhibits the Warburg effect and cell proliferation through the Wnt/β-catenin pathway, leading to a better prognosis in CRC [122].

### 4.2. SLC56 Family

The SLC56 family, also known as sideroflexins (SFXN), contains five homologs: SFXN1, SFXN2, SFXN3, SFXN4, and SFXN5. All SFXN members are highly conserved mitochondrial transmembrane proteins, whose N-terminus and C-terminus have the same topological structure [123]. Recently, it was discovered that SFXN1 is a serine transporter [124]. SFXN1 and SFXN3 are highly homologs. SFXN1 is a transporter for serine, alanine, glycine, and cysteine [123]. As the main source of single carbon units for biosynthesis, serine can be catabolic into glycine by serine hydroxymethyltransferase 2 (SHMT2) in the mitochondria. Therefore, SFXN1 closely regulates serine and glycine levels and alanine synthesis in the mitochondria. SXFN1 is strongly associated with lung cancer. Compared with paracancerous tissues, the mRNA expression of SFXN1 is significantly increased in cancer tissues, and SFXN1 promotes the progression of NSCLC by activating the mTOR signaling pathway [125].

### 4.3. MPC Family

The mitochondrial shuttle process of pyruvate connects glycolysis and oxidative phosphorylation. In normal cells, pyruvate is mostly produced by glycolysis, and the cytoplasmic metabolism of pyruvate is controlled by the cellular microenvironment. Pyruvate is usually converted to pyruvate and transported to extracellular space under hypoxia. In the presence of sufficient oxygen, pyruvate is transported to the mitochondrial matrix by the mitochondrial pyruvate carrier and oxidized to Acetyl-CoA to participate in the TCA cycle. MPC belongs to the SLC54 family. Human MPC consists of the MPC1 and MPC2 dimer [126]. Loss of one MPC subunit leads to the degradation of the other MPC subunit, preventing pyruvate transportation to the mitochondria [127]. Unlike normal cells, pyruvate oxidation is often restricted in cancer cells because of downregulated mitochondrial carriers of pyruvate. In fact, both of these transporters, particularly MPC1, are downregulated or absent in most cancer cells, and low MPC expression is associated with poor survival. MPC1 overexpression in renal cell carcinoma inhibited tumor growth and invasion in vivo [128]. Increased expression of MPC1 restored the mitochondrial metabolism of pyruvate and inhibited glycolysis. Therefore, MPC1 is considered a tumor suppressor gene.

MPC1 affects cancer progression by regulating tumorigenicity, tumor stemness, and chemoresistance [129,130,131]. Several molecules regulate MPC1 to interfere with cancer cell proliferation. Lysine demethylase 5A (KDM5A), a substance originally identified as a retinoblastoma-binding protein, regulates MPC1 expression by demethylating lysine 4 of histone H3 (H3K4) at the transcriptional level in pancreatic ductal cancer cells [132]. In hepatocellular carcinoma cells, PGC1α forms a complex with NRF1 and binds to the MPC1 promoter, ultimately increasing ROS formation and inducing apoptosis of HCC cells [133].

Up to now, several MPC inhibitors have been discovered. UK-5099 is a standard small-molecule MPC inhibitor that is specific for MPC at low concentrations and is often used as an instrumental drug in basic research [134]. Zaprinast is a specific MPC inhibitor that inhibits pyruvate-driven O_2_ consumption in brain mitochondria and blocks MPC in liver mitochondria [135]. 7ACC2 is also a potent MPC inhibitor that can continuously block extracellular lactate uptake by promoting intracellular pyruvate accumulation [136]. In addition, Wesley, T. et al. found through the pharmacophore model that carsalam, six quinolone antibiotics, and 7ACC1 and 7ACC2 shared pharmacophore, which was a new MPC inhibitor [137] (Figure 3).

### 4.4. Others

Han et al. revealed that SLC1A5 (ASCT2) transports glutamine to the mitochondria and plays an important role in cancer metabolism [138]. SLC1A5 is involved in the regulation of many oncogenic signaling pathways. Studies have shown that STAT3 regulates MYC expression in acute myeloid leukemia (AML), thereby controlling SLC1A5 transcription and OXPHOS and promoting the survival of leukemia stem cells [139]. Other studies have shown that neural precursor cells inhibit autophagy and mitochondrial metabolism by downregulating Unc51-like kinase 1 (ULK1) or ASCT2, thereby inhibiting the growth and survival of pancreatic cancer cells [138]. In addition, SLC1A5 expression was positively correlated with the number of tumor-infiltrating B cells, CD4^+^ T cells, CD8^+^ T cells, macrophages, neutrophils, and dendritic cells in HCC and low-grade glioma (LGG), indicating its role in regulating tumor immunity [140]. In conclusion, SLC1A5 is a key target for cancer therapy.

## 5. Conclusions

Recent findings on metabolic reprogramming highlight the importance of mitochondria. Mitochondria are involved in metabolic reprogramming, while the particularity of its structure and the diversity of its functions hinder the development of targeted drugs. The discovery of mitochondria-specific transporters supports mitochondrial phenotypic changes in tumor cells and provides new opportunities for developing new anticancer drugs.

## Figures and Tables

**Figure 1 ijms-24-04954-f001:**
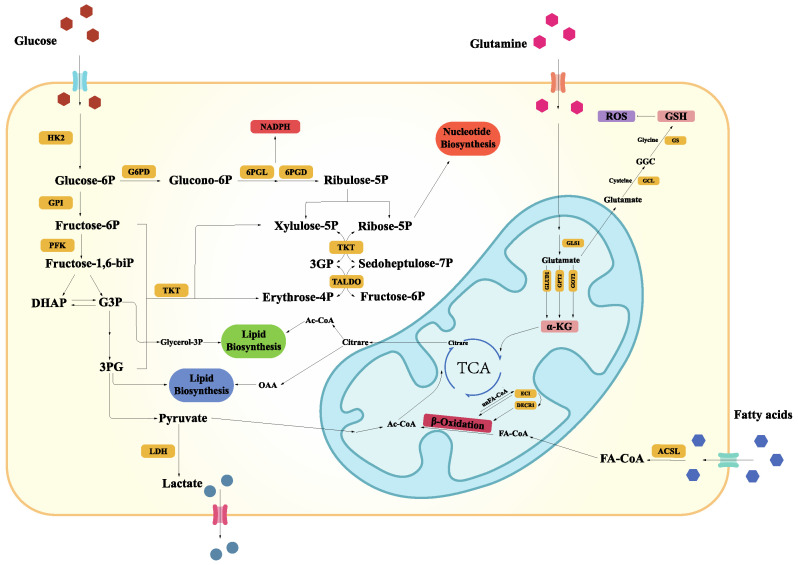
The metabolic reprogramming involved in mitochondria promotes the occurrence and progression of tumors. Tumor cells need more nutrients to meet the demand for survival and proliferation. Under aerobic conditions, glucose as the primary energy source is firstly converted to pyruvate by glycolysis. Pyruvate is used in lactate production, can also be used in other biosynthetic reactions, or is transported to the mitochondrial matrix to participate in the TCA cycle. As energy supplements, glutamine and fatty acids are also transported into mitochondria to take part in the TCA cycle in the form of α-KG and Acetyl-CoA, respectively, for synthesizing biological macromolecules such as nucleotides, amino acids, and lipids. (α-KG, α-ketoglutaric acid; ACSL, Long-chain Acyl-CoA Synthetase; DHAP, dihydroxyacetone phosphate; DECR1, 2,4-dienoyl CoA reductase 1; ECI, △3-△2-enoyl-CoA isomerase; G6PD, Glucose-6-phosphate Deplrdrogenase; GSH, glutathione; GPI, glucose-6-phosphate isomerase; GS, glutathiol; GCL, Glutamate cysteine ligase; GLS, glutaminase; GLUD1, glutamate dehydrogenase 1; GPT2, glutamate pyruvate transaminase 2; GOT2, glutamate oxaloacetate transaminase 2; HK2, hexokinase-2; LDH, lactate dehydrogenase; OAA, oxaloacetate; PFK, phosphofructokinase; 6PGD, 6-phosphogluoconate dehydrogenase; 6PGL, 6-phosphogluconolactonase; PFK, phosphofructokinase; ROS, reactive oxygen species; TKT, transketolase; TALDO, transaldolase).

**Figure 2 ijms-24-04954-f002:**
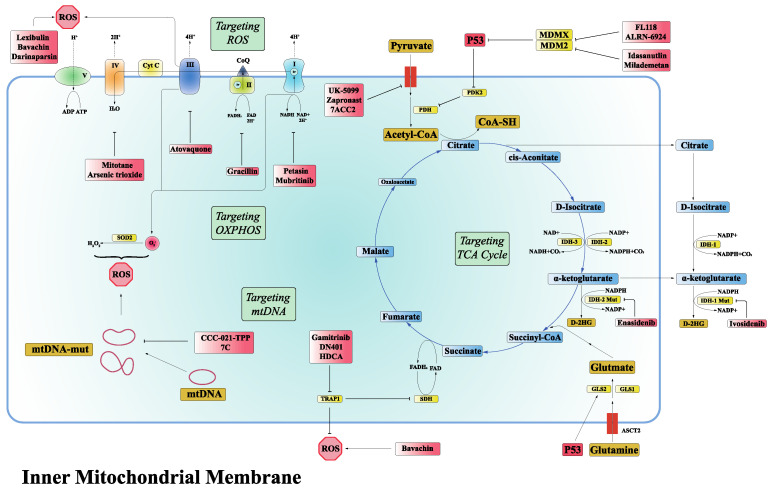
Therapeutic strategies targeting mitochondrial metabolic reprogramming. According to the functional characteristics of mitochondria, current therapies targeting mitochondria can be divided into targeting the TCA cycle, OXPHOS, ROS, and mtDNA. Small molecule compounds developed for the above targets can precisely inhibit the occurrence and progression of cancer by interdicting the normal functioning of mitochondria. (D-2HG, D-2 hydroxy-glutarate; GLS2, glutaminase 2; GLS1, glutaminase 1; HDCA, Honokiol bis-dichloroacetate; IDH, isocitrate dehydrogenase; NAD^+^, nicotinamide adenine dinucleotide; NADH, nicotinamide adenine dinucleotide; OXPHOS, oxidative phosphorylation; PDH, pyruvate dehydrogenase; PDK2, pyruvate dehydrogenase kinase 2; ROS, reactive oxygen species; SDH, succinate dehydrogenase; TRAP1, Tumor necrosis factor receptor-associated protein 1.).

**Figure 3 ijms-24-04954-f003:**
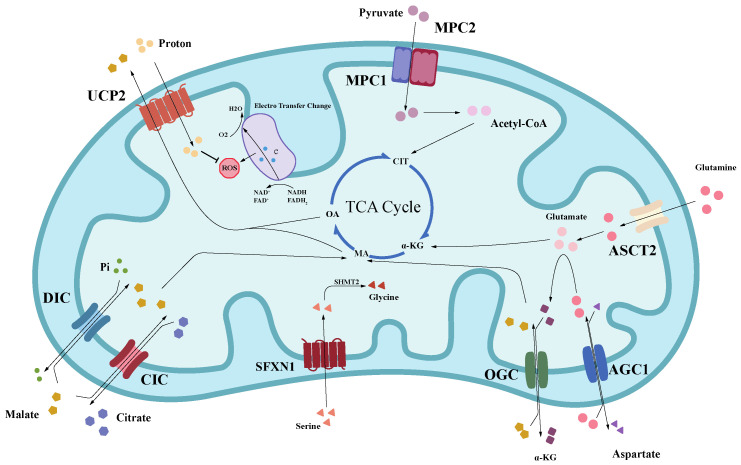
The transporters located in the IMM are ideal targets for anticancer drugs. The transporters located in the inner membrane of mitochondria act as the essential hub connecting the mitochondrial matrix with the cytoplasmic matrix. Targeting transporters with abnormal expression in tumor cells to play an anticancer role has achieved the expected results in both in vitro and in vivo experiments. (ASCT2, alanine/serine/cysteine transporter 2; α-KG, α-ketoglutaric acid; AGC1, aspartate/glutamate carrier 1; CIT, citrate; MA, malic acid; OA, oxaloacetic acid; CIC, citrate carrier; DIC, dicarboxylate carrier; FAD^+^, flavin adenine dinucleotide; FADH_2_, dihydroflavine-adenine dinucleotide; MPC, mitochondrial pyruvate carrier; NAD^+^, nicotinamide adenine dinucleotide; NADH, nicotinamide adenine dinucleotide; OGC, oxoglutarate carrier; ROS, reactive oxygen species; SHMT2, hydroxymethyltransferase 2; SFXN1, sideroflexins 1; UCP2, uncoupling protein 2).

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
