# Peer review of "Targeting Mitochondrial Metabolic Reprogramming as a Potential Approach for Cancer Therapy"

_ijms, 2023, doi:10.3390/ijms24054954_

Round 1

Reviewer 1 Report

The authors present a general overview of metabolic reprogramming in cancer cells. The manuscript is well presented. I have some suggestions:

I would suggest to add some information on the specific complex I inhibition found in several toumors like ovarian cancer and comment it. 

The Legends to figures are few esplicative. It is not immediate to understand what changes with respect to normal cells. Please add more information on the molecular mechanisms in the legends in order to make the figure more impactful. 

Figure 2 needs to be bigger. The complex III pumps four protons, please correct in the figure 2. 

The full name of some acronyms, such as LDH or MPC, should be added to the text and not only in the legends to figures. 

Author Response

Thank you very much for your kind comments on February 03, 2023. The manuscript has been revised carefully. In order to present the manuscript clearly, some parts have been reorganized and rewritten. As to English, we have checked and modified the whole manuscript carefully to avoid language errors and improve the quality of the manuscript. All the changes in the revised manuscript have been clearly indicated with a track changes. The comments and our responses are highlighted as follows:

Reviewer #1’s comments:

The authors present a general overview of metabolic reprogramming in cancer cells. The manuscript is well presented. I have some suggestions:

Comment 1: I would suggest to add some information on the specific complex I inhibition found in several toumors like ovarian cancer and comment it.

Author’s response: Thank you for your corrections. We concerned that targeting complex I is becoming a hot topic in the field, especially recently for IACS-010759. Therefore, we summarized the development of related drugs, and expressed our views at the end of this section. The discussion about complex I has been described as “Complex I is located at the frontline of the respiratory chain. As a major producer of proton gradients in ETC, complex I is a suitable target for the development of an OXPHOS inhibitor. As a marketed drug, metformin has received much attention for its ability to inhibit complex I, but its low potency has limited the potential for re-purposing [60]. BAY87-2243 was also promising, but severe vomiting in phase I trials prevented its further development. In recent years, more and more efficient and selective small-molecule drugs have been developed [61,62]. EVT-701, a novel small-molecule inhibitor targeting diffuse large B-cell lymphoma and NSCLC, has demonstrated good efficacy in vitro and in vivo. It should be emphasized that the original structure of BAY87-2243 was deliberately modified to prevent severe side effects in clinical trials” and “Recent clinical studies have shown that targeting OXPHOS is not very feasible. IACS-010759, a small molecule inhibitor targeting complex I, had a narrow therapeutic index and serious side effects such as elevated lactic acid level and neurotoxicity in a phase I clinical study [89]. Therefore, it is necessary to monitor the toxicity of anti-tumor drugs targeting complex I”.

Comment 2: The Legends to figures are few esplicative. It is not immediate to understand what changes with respect to normal cells. Please add more information on the molecular mechanisms in the legends in order to make the figure more impactful.

Author’s response: Thank you for your corrections. Each illustration has been properly explained to make readers' understanding of the text more accurate. For example, we have summarized the molecular principles of the first figure to make it more attractive to the reader. The information for figure 1 has been described as “Tumor cells need more nutrients to meet the demand for survival and proliferation. Under aero-bic conditions, glucose as the primary energy source is firstly converted to pyruvate by glycolysis. Pyruvate is used in lactate production, can also be used in other biosynthetic reactions, or is transported to the mitochondrial matrix to participate in the TCA cycle. As energy supplements, glutamine and fatty acids are also transported into mitochondria to take part in the TCA cycle in the form of α-KG and Acetyl-CoA, respectively, for synthesizing biological macromolecules such as nucleotides, amino acids, and lipids”.

Comment 3: Figure 2 needs to be bigger. The complex III pumps four protons, please correct in the figure 2.

Author’s response: Thank you for your advice. We have enlarged the illustration elements as much as possible and would like to provide a more explicit version. The typo mentioned above has been corrected to ensure accurate representation.

Comment 4: The full name of some acronyms, such as LDH or MPC, should be added to the text and not only in the legends to figures.

Author’s response: Thank you for your advice. We have corrected this typo mentioned above to make our text fit for submission.

Reviewer 2 Report

This review by the Drs. Zhang et al. discusses the latest findings in targeting mitochondrial metabolic reprogramming as a potential approach for cancer therapy. This review is comprehensive, very well organized and written; and represents summary of a field with great importance.

I have just a couple of minor remarks, please check:

Abstract,p.1 r7- word missing after “mitochondrial…” ?

p.1 r9- “biological energy” –please rephrase

p.1 r22- “scholars”?-please replace

p.1 r23-24- “makes… become” –please correct English

Author Response

Thank you very much for your kind comments on February 07, 2023. The manuscript has been revised carefully. In order to present the manuscript clearly, some parts have been reorganized and rewritten. As to English, we have checked and modified the whole manuscript carefully to avoid language errors and improve the quality of the manuscript. All the changes in the revised manuscript have been clearly indicated with a track changes. The comments and our responses are highlighted as follows.

Reviewer #2’s comments:

This review by the Drs. Zhang et al. discusses the latest findings in targeting mitochondrial metabolic reprogramming as a potential approach for cancer therapy. This review is comprehensive, very well organized and written; and represents summary of a field with great importance.

I have just a couple of minor remarks, please check:

Comment 1: Abstract, p.1 r7- word missing after “mitochondrial…”?

Author’s response: Thank you for your corrections. We have corrected this typo to make our text fit for submission. The “mitochondrial” has been changed to “mitochondria”, as follows: “Abnormal energy metabolism is a characteristic of tumor cells, and mitochondria are important components of tumor metabolic reprogramming”.

Comment 2: p.1 r9- “biological energy” –please rephrase

Author’s response: Thank you for your corrections. We have made a comprehensive correction to the content of the Abstract section. The “biological energy” has been changed to “chemical energy”, as follows: “Mitochondria have gradually received the attention of scientists due to their important functions such as providing chemical energy, producing substrates for tumor anabolism, controlling RE-DOX and calcium homeostasis, participating in the regulation of transcription, and controlling cell death”.

Comment 3: p.1 r22- “scholars” ? -please replace

Author’s response: Thank you for your advice. We have corrected this typo to ensure accurate representation. The sentence “scholars began to regard cancer as a mitochondrial metabolic disease” has been changed to “Previous studies have shown that cancer is a mitochondrial metabolic disease”.

Comment 4: p.1 r23-24- “makes… become” –please correct English

Author’s response: Thank you for your advice. We have optimized the expression of our sentences so that readers can better understand what we mean. We have changed “the Warburg effect makes aerobic glycolysis become the main source of energy for cancer cells” to “the Warburg effect makes aerobic glycolysis the main source of energy for cancer cells”.
